# MODA: MOdular Duplex Attention for
# Multimodal Perception, Cognition, and Emotion Understanding

**Zhicheng Zhang**[1 2 †]  **Wuyou Xia**[1]  **Chenxi Zhao**[1 †]  **Yan Zhou**[3]  **Xiaoqiang Liu**[3]  **Yongjie Zhu**[3 ‡]  **Wenyu Qin**[3]
**Pengfei Wan**[3]  **Di Zhang**[3]  **Jufeng Yang**[1 2]

## Abstract

Multimodal large language models (MLLMs) recently showed strong capacity in integrating data among multiple modalities, empowered by a generalizable attention architecture. Advanced methods predominantly focus on language-centric tuning while less exploring multimodal tokens mixed through attention, posing challenges in high-level tasks that require fine-grained cognition and emotion understanding. In this work, we identify the attention deficit disorder problem in multimodal learning, caused by inconsistent cross-modal attention and layer-by-layer decayed attention activation. To address this, we propose a novel attention mechanism, termed MOdular Duplex Attention (MODA), simultaneously conducting the inner-modal refinement and inter-modal interaction. MODA employs a correct-after-align strategy to effectively decouple modality alignment from cross-layer token mixing. In the alignment phase, tokens are mapped to duplex modality spaces based on the basis vectors, enabling the interaction between visual and language modality. Further, the correctness of attention scores is ensured through adaptive masked attention, which enhances the model's flexibility by allowing customizable masking patterns for different modalities. Extensive experiments on 21 benchmark datasets verify the effectiveness of MODA in perception, cognition, and emotion tasks. ***Source code and demo are available in https://zzcheng.top/MODA.***

---

[†]Work done during the internship at Kuaishou [‡]Project Leader
[1]VCIP & TMCC & DISSec, College of Computer Science, Nankai University [2]Pengcheng Laboratory [3]Kuaishou Technology. Correspondence to: Jufeng Yang <yangjufeng@nankai.edu.cn>.

*Proceedings of the 42$^{nd}$ International Conference on Machine Learning*, Vancouver, Canada. PMLR 267, 2025. Copyright 2025 by the author(s).

**(a)**

**(b)**

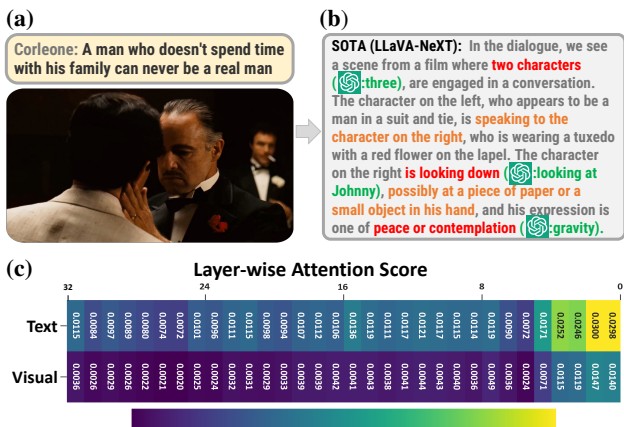

*Figure 1.* **Illustration of deficit disorder attention problem.** (a) Given the detailed image and lines from *The Godfather*, (b) we highlight incorrect responses, corresponding hallucinated explanations, and attached answers. (c) We visualize attention score across layers, highlighting inconsistent attention across modalities.

## 1. Introduction

Benefiting from the blossom of large language models (Chiang et al., 2023; Dubey et al., 2024; Teknium et al., 2024), multimodal large language models (MLLMs) have shown strong capacity in integrating multimodal data as human (Tong et al., 2024a; Brown et al., 2020; Bai et al., 2023), which illuminate a promising pathway toward Artificial General Intelligence (AGI). Advanced effort has been devoted to constructing MLLM (Achiam et al., 2023), focusing on exploring more insightful data curation, model tuning, and evaluation benchmarks. As the controller of agent, MLLMs provide a natural solution by conducting content perception (Liu et al., 2023), understanding role cognition (Dai et al., 2025), and analyzing human emotion (Yang et al., 2024). One more step forward into AGI lies in high-level multimodal understanding like humans, including cognition and emotion. Cognition, as a higher-level capability, requires the ability to model relationships and reasoning across modalities (Dai et al., 2025; Pessoa, 2022). Beyond cognition, emotion understanding is another critical aspect of fine-grained multimodal comprehension (Yang et al., 2024; Zhang et al., 2023). These high-level multimodal tasks pose new challenges for MLLMs.

While recent MLLMs show promising results in basic perception, they still struggle to perceive fine-grained details (Tong et al., 2024b), which is essential for understanding cognition and emotion. Public benchmarks reveal that these advanced MLLMs can underperform relative to random guessing (Yang et al., 2024), with 3 SOTAs achieving approximately 50:50 accuracy in 2-class sarcasm detection on the HFM dataset. This phenomenon arises from an excessive emphasis on the dominant modality data, leading to neglect of fine-grained details in alternative modality.

We delve deep into the reason and analyze the multimodal tokens mixed by attention in MLLM. As shown in Fig. 1 (a)&(b), we observe that SOTA MLLM struggles to capture fine-grained details (*e.g.*, eyesights of character), leading to error in emotion understanding. The reason behind this is inconsistent attention across multiple layers in MLLM (63% disparity in Fig. 1 (c)), which we call deficit disorder attention problem. On the one hand, the attention scores in MLLM exhibit a bias towards the language component. On the other hand, layer-by-layer decay of attention further accentuates this disparity. As a result, the attention score disparity across modalities can reach up to 10 times.

Our intuition is that multimodal attention mechanisms often suffer from imbalances between self-modal and cross-modal interactions, leading to suboptimal feature co-operation across modalities. By explicitly separating and modulating these two components, we can better align multimodal features while preserving the unique characteristics of each modality. To achieve this, we propose MOdular Duplex Attention (MODA), which splits attention into self-modal and cross-modal parts, each with its own modulated attention mask. The self-modal attention component focuses on capturing the intrinsic relationships within individual modalities. In contrast, the cross-modal attention component is responsible for aligning features across different modalities, facilitating effective information exchange. At the core of the MODA model is the Duplex (V/T)-Aligner, which maps the tokens into a shared dual-modality representation space defined by two gram matrices. Additionally, the Modular Masked Attention component allows the model to adaptively focus on relevant modalities by applying customized masking patterns, further enhancing its flexibility on multimodal understanding tasks.

Our contributions are two-fold as follows: (1) From a novel perspective of the attention shift mechanism, we indicate the key bottleneck of attention among SOTA MLLMs and analyze the core reason in depth. We further propose a modular and duplex attention mechanism based on our observation. (2) We investigate a new MLLM for perception, cognition, and emotion, enabling applications in fine-grained understanding and planning. Extensive experiments on 21 benchmarks verify the generalization and effectiveness of MODA.

## 2. Related Work

**Multimodal large language model (MLLM)** have garnered significant attention recently due to their ability to integrate pre-trained foundational models, especially powerful Large Language Models (LLMs)(Achiam et al., 2023; Touvron et al., 2023), alongside multimodal encoders(Dosovitskiy et al., 2021; Radford et al., 2021). These models enhance the processing of multimodal inputs and outputs, as demonstrated in advanced works (Alayrac et al., 2022; Bai et al., 2023). MLLMs leverage attention mechanisms to facilitate multimodal token mixing, enabling both inductive and deductive understanding across modalities. However, the vision modality's potential remains underutilized in many of these models. MMVP (Tong et al., 2024b) identifies a critical issue, highlighting how existing MLLMs fail to fully activate the vision modality due to improper handling of low-level visual attributes. Further, Cambrian-1 (Tong et al., 2024a) confirms this limitation and introduces a spatial vision aggregator to enhance visual feature. In this work, we investigate the root cause of these limitations, identifying the bottleneck in the design of the multimodal attention mechanism. To address the issue of imbalanced attention scores, we propose a novel multimodal attention that better balances the contributions of each modality.

**Understanding cognition and emotion** (Fu et al., 2023; Yang et al., 2024) play an important role in the pathway toward building an intelligent agent, except for content understanding demonstrated by prior MLLMs. As two of high-level understanding, cognition (Wang et al., 2024a; Kong et al., 2024; Salemi et al., 2024) typically refers to the ability to make decisions and judgments similar to characters (Binz & Schulz, 2023; Wang et al., 2024c; Deshpande et al., 2023), such as generating website code (Zhu et al., 2024; Wang et al., 2025), or role playing (Chen et al., 2024; Zhang et al., 2018). Emotion mainly depends on the psychology assumptions (Zhao et al., 2021; Zhang et al., 2024), where the categorical one is mostly used due to it being easily understandable (Yang et al., 2018; Mai et al., 2022; Lian et al., 2022; Zhang & Yang, 2022). However, it is less explored due to its requirements for fine-grained content understanding, which MLLMs can hardly achieve.

**Attention in MLLM** plays a critical role in addressing the computational and memory challenges inherent in their design. Significant progress has been made in developing efficient attention mechanisms for Transformer architectures, which include fixed patterns (Child et al., 2019), combinations of patterns (Zaheer et al., 2020), learnable patterns (Kitaev et al., 2020), neural memory (Beltagy et al., 2020), low-rank methods (Wang et al., 2020), and kernel-based techniques (Choromanski et al., 2021). For example, the Set Transformer introduces inducing points to handle set-input problems (Wang et al., 2020), while the Axial

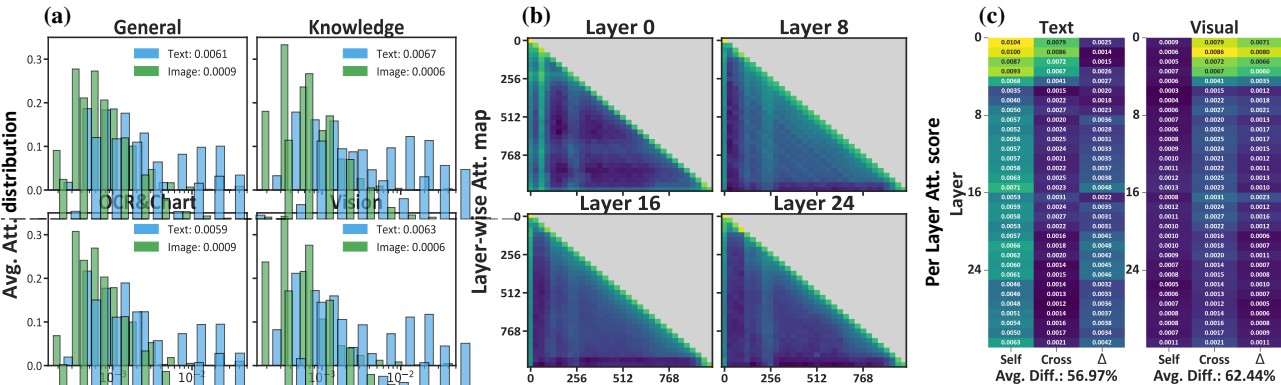

*Figure 2.* Analysis of existing MLLMs on four fine-grained understanding tasks. (a) The distribution of attention activation values among visual and textual tokens. (b) The attention map for multimodal tokens among stages. (c) The self- and cross-modal attention activation scores with their disparity among the attention layers.

Transformer applies attention along individual axes of input tensors, reducing computational overhead (Beltagy et al., 2020). These innovations collectively enhance the scalability of Transformer models, enabling their application to tasks with large inputs or long sequences (Choromanski et al., 2021; Han et al., 2024). While previous approaches have focused on improving the efficiency and scalability of attention in single-modal tasks, the multimodal context introduces unique challenges, such as balancing attention scores across heterogeneous modalities (Zhao et al., 2021). Our work extends this line of research by specifically addressing the multimodal attention mechanism in MLLMs.

## 3. Methodology

### 3.1. Preliminary

• **Attention**  Given the input multimodal tokens, $X \in \mathbb{R}^{N \times d}$, $N$ be the number of tokens and $d$ be the dimensionality of the hidden state. Let $A \in \mathbb{R}^{N \times N}$ denote the attention score matrix computed among $N$ tokens, we have $A = QK^\top/\tau$, and the output of attention layer as:

$$O = \text{Softmax}(\frac{QK^\top}{\tau} + M)V. \quad (1)$$

where $Q, K, V \in \mathbb{R}^{d \times d}$ represents query, key, and value matrix derived from input tokens. Attention is also practically masked $M \in \mathbb{R}^{N \times N}$ to filter out special tokens (Li et al., 2023) or conduct causal sequential modeling (Wang et al., 2024b; Achiam et al., 2023).

• **Multimodal Attention**  Formally, consider a multimodal token sequence $X_M$ comprising $M$ modalities. The total token length is $N_M = N_1 + \cdots + N_M$, where $N_m$ represents the length of the $m^{th}$ modality token sequence $X_m$. The attention can be split into two parts for each modality token

sequence, self-modal attention and cross-modal attention. We have $(\cdot)^{[m,\bar{m}]}$, which represents the tokens derived from the $m^{th}$ modality and rest. For the self-modal and cross-modal attention, we have

$$O_{self} = \text{Softmax}(\frac{Q^m K^{m\top}}{\tau} + M)V^m, \quad (2)$$

$$O_{cross} = \text{Softmax}(\frac{Q^m K^{\bar{m}\top}}{\tau} + M)V^{\bar{m}}. \quad (3)$$

### 3.2. Deficit Disorder Attention Problem

Recently, multimodal attention has played a very important role in multimodal areas, including diffusion models that involve cross-modal generation and MLLM that involves cross-modal understanding. The attention mechanism governs token interactions by computing similarities and applying masks. To further investigate the Attention Deficit Disorder (DDA) phenomenon, we conduct a series of analyses on four categories of fine-grained understanding tasks.

As shown in Fig. 2 (a), we observe that the attention devoted to visual content is significantly weaker compared to that for the textual modality. This observation aligns with the challenges faced by MLLMs fine-tuned from autoregressive models in handling fine-grained visual perception. The inherent design of MLLM, which is primarily optimized for text-based tasks, may lead to an underrepresentation of visual features when extended to multimodal contexts. This imbalance highlights a critical limitation in the current architecture, where the model's proficiency in textual processing does not seamlessly translate to an equivalent capability. Further, we conduct experiments on Fig. 2 (b)&(c), and we observe a distinct cross-attention bias in the lower layers of the model across its 32 layers. This bias is notably inconsistent with the distribution of attention in the higher layers, which are known for their stronger representational capabilities. Specifically, the lower layers tend to focus

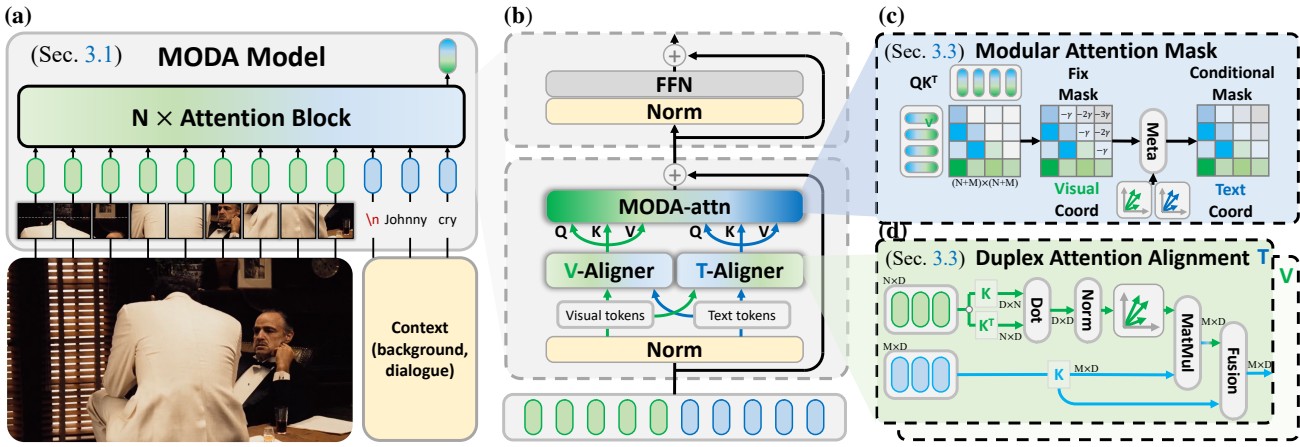

*Figure 3.* MOdular Duplex Attention. (a) MODA takes the image and contextual prompt as input, including the background and history of the conversation. (b) With MODA, the token flows are justified in each Transformer block of MLLM. MODA modifies the deficient attention scores in a correct-after-align manner via (c) Modular masked attention and (d) Duplex (V/T)-aligner.

disproportionately on cross-modal interactions, potentially at the expense of effectively capturing intra-modal features, leading to suboptimal multimodal integration.

This leads to the formal introduction of the Deficit Disorder Attention (DDA) problem. Given the visual tokens $x_v^l$ and text tokens $x_t^l$ in the block $l$, the multimodal attention builds the link from two parts (*i.e.*, self-modal $x_t^l \rightarrow x_t^{l+1}$, $x_v^l \rightarrow x_v^{l+1}$ and cross-modal $x_t^l \rightarrow x_v^{l+1}$, $x_v^l \rightarrow x_t^{l+1}$ ), where the links are commonly implemented by the pair-wise token similarity and weighted sum. However, the modality gap between tokens decrease the magnitude of links, as we observed, the link value of $x_v^l \rightarrow x_v^{l+1}$ and $x_t^l \rightarrow x_v^{l+1}$ decays exponentially with depth ($\alpha_{v,t \rightarrow v}^l \propto \gamma^l, \gamma \neq 1$). This misalignment propagates layer-wise, causing the cumulative error in cross-modal interaction to grow as

$$\mathbb{E}_{DDA} = \prod_l \gamma^l \epsilon_l, \tag{4}$$

where $\epsilon_l$ denotes the layer-specific alignment error. This phenomenon aligns with the theoretical insights in (Dong et al., 2021), where pure attention mechanisms experience rank collapse, a critical factor that exacerbates the imbalance in attention distribution.

### 3.3. MOdular Duplex Attention

When the gap across modalities arises, we propose to align the tokens from multiple modalities in the attention, which we call modular duplex attention (MODA). MODA first splits multimodal attention into the modality alignment part and the token focus correction part.

● **Duplex Attention Alignment** To reduce the modality inconsistency, a natural idea is to align them. Inspired by the recent advance of visual-language embedding space map-

ping in diffusion model (Rombach et al., 2022), we propose mapping the token into the other modality space according to the embedding space bases of the gram matrix. We extract the basis vector of each modality space according to the gram matrix of tokens (Ryu et al., 2023; Peebles & Xie, 2023), thus compressing the semantics of each modality and serving as a transfer for other modalities. Thus, the duplex attention alignment consists of V-Aligner and T-Aligner responsible for visual and language modality, respectively.

Specifically, for the $m^{th}$ modality, the space bases are given according to the normed gram matrix $||\boldsymbol{G}^m|| \in \mathbb{R}^{d \times d}$, where $\boldsymbol{G}_{ij}^m$ is the inner product between tokens $i$ and $j$:

$$\boldsymbol{G}_{ij}^m = \sum_{k=1}^{N_m} \boldsymbol{K}_{ik}^m \boldsymbol{K}_{kj}^m = \boldsymbol{K}^{m\top} \boldsymbol{K}^m, \tag{5}$$

where $\boldsymbol{K}^m$ are the key states of the $m^{th}$ modality tokens and $N_m$ is the number of token belong to modality $m$. By including the base vectors of the space defined by the Gram matrix, we can effectively capture the relationships among the tokens within the $m^{th}$ modality. This allows us to construct a feature representation that is not only rich in information but also maintains the intrinsic structure of the data.

As a following product, the normed gram matrix serves as a cross-modal token transfer function, enabling an efficient transformation of tokens from other modality $\bar{m}$ into the modality $m$ as a kernelized mapping function $f : \mathbb{R}^d \rightarrow \mathbb{R}^d$. The aligned tokens are computed as follows:

$$\boldsymbol{K}^{\bar{m} \rightarrow m} = \boldsymbol{K}^{\bar{m}} ||\boldsymbol{G}^m||, \tag{6}$$

where $\boldsymbol{K}^{\bar{m}}$ represents the value representation from other modalities $\bar{m}$. The mapped tokens are further fused with the original ones to enhance the token similarity among

*Table 1.* **Ablation Study.** We conduct experiments on four types of multimodal tasks, including general QA (**G**), knowledge QA (**K**), OCR&Chart QA (**O**), and vision-centric QA (**V**). The lines with blue shallow indicate the optimal setting for our method. If not otherwise specified, this setting is used for all subsequent experiments.

| (a) **Module** | | | | | | (b) **Attention Alignment** | | | | | (c) **Attention Fusion** | | | | | (d) **Attention Mask** | | | | |
|---|---|---|---|---|---|---|---|---|---|---|---|---|---|---|---|---|---|---|---|---|
| MDM | DAA | **G** | **K** | **O** | **V** | align | **G** | **K** | **O** | **V** | fusion | **G** | **K** | **O** | **V** | mask | **G** | **K** | **O** | **V** |
| - | - | 63.6 | 44.0 | 60.8 | 38.0 | MLP | 69.5 | 47.5 | 66.8 | 46.0 | $X_p$ | 69.2 | 45.4 | 60.9 | 42.6 | Inf | 67.8 | 47.6 | 63.3 | 48.1 |
| ✓ | - | 69.2 | 45.4 | 60.9 | 42.6 | +2xMLP | 66.5 | 48.6 | 67.9 | 49.1 | $X_a$ | 67.8 | 47.6 | 63.3 | 48.1 | Fix | 70.1 | 49.0 | 67.0 | 52.3 |
| - | ✓ | 67.8 | 47.6 | 63.3 | 48.1 | +GeLU | 69.5 | 49.1 | 64.0 | 54.2 | Con | 69.3 | 48.3 | 67.0 | 54.3 | Attn. | 69.3 | 48.3 | 67.0 | 54.3 |
| ✓ | ✓ | 69.3 | 48.3 | 67.0 | 54.3 | +CoV | 69.3 | 48.3 | 67.0 | 54.3 | Add | 62.2 | 47.6 | 67.2 | 52.2 | [M] | 69.5 | 47.5 | 66.8 | 46.0 |

all the modalities. Due to the substantial computational expense associated with training a complete MLLM, we utilize token merging and LoRA-based tuning to develop the fuser. Notably, the computation in the alignment stage keeps linear complexity to the token number, since the matrix sum among tokens is only conducted in the first round.

● **Modular Attention Mask** Attention mask controls the flow of tokens across transformer layers and induces the positional bias for MLLM (Wu et al., 2024). To better fit the requirements of the multimodal token sequence, we assign a modulated attention mask for each modality, where the mask is split into $M^m$ and $M^{\bar{m}}$ responsible for self- and cross-modality, respectively.

$$O_{self} = \text{Softmax}(\frac{Q^m K^{m\top}}{\tau} + M^m)V^m, \quad (7)$$

$$O_{cross} = \text{Softmax}(\frac{Q^m K^{\bar{m}\top}}{\tau} + M^{\bar{m}})V^{\bar{m}}. \quad (8)$$

To alleviate the collapsed attention matrix and prevent it from under-smoothed tokens. We first introduce a modular attention mask that stores unnecessary attention values as pseudo-attention scores (Yin et al., 2024). For each row, representing the attention scores for the $i$-th token, the sequence length that the token can attend to is fixed at $n$. Consequently, each row contains $n - i$ pseudo-attention scores, which are allocated to the excess values. The attention scores are formally represented using a masking strategy with a decay rate $\beta$, as follows:

$$A_{MM} = \begin{pmatrix} q_1 k_1^\top & p_{11} & \cdots & p_{1(n-1)} \\ q_2 k_1^\top & q_2 k_2^\top & \cdots & p_{1(n-2)} \\ \vdots & \vdots & \ddots & \vdots \\ q_n k_1^\top & q_n k_2^\top & \cdots & q_n k_n^\top \end{pmatrix} \quad (9)$$

$$p_{base} = 0, p_{ij} = p_{base} - (j-1)\beta \quad (10)$$

Except for the absolute location prior information, we further introduce the modality location to enforce the model to correct the token flow. We introduce the normed gram matrix as an indicator, to find out the part should be carried with modality location priors. We introduce the normed Gram

matrix to serve as a critical indicator, guiding the model in identifying which components should leverage modality location priors. This separation allows for more precise control over how tokens from the same modality interact with each other versus how they engage with tokens from other modalities. The self-modal attention, represented by $O_{self}$, focuses on refining the relationships within the same modality, ensuring that relevant information is effectively propagated through the layers. Conversely, the cross-modal attention, denoted by $O_{cross}$, facilitates the exchange of information between distinct modalities, enabling the model to leverage complementary features.

## 4. Experiment

### 4.1. Benchmark Datasets

***Perception***: Following (Tong et al., 2024a), we conduct experiments on 4 types of perception task (*i.e.*, general, knowledge, ocr, and vision-centric) across 16 benchmarks: MME (Fu et al., 2023), MMBench (Liu et al., 2025), SEED (Li et al., 2024), GQA (Hudson & Manning, 2019), ScienceQA (Lu et al., 2022), MMMU (Yue et al., 2024), MathVista (Lu et al., 2024), AI2D (Kembhavi et al., 2016), ChartQA (Masry et al., 2022), OCRBench (Liu et al., 2024), TextVQA (Singh et al., 2019), DocVQA (Mathew et al., 2021), MMVP (Tong et al., 2024b), RealworldQA (xAI, 2024), and CV-Bench (Tong et al., 2024a). We adopt GPT4 score to evaluate response.

***Cognition***: Following (Dai et al., 2025), we conduct experiments on MMRole to evaluate role-playing performance from 8 aspects: instruction adherence, fluency, coherency, image-text relevance, response accuracy, personality consistency, knowledge consistency, and tone consistency.

***Emotion***: Following (Yang et al., 2023; Huang et al., 2024), we conduct experiments on 4 benchmark datasets. MVSA-S and MVSA-M (Niu et al., 2016) are datasets used for sentiment polarity classification (positive or negative), while TumEmo (Yang et al., 2021) is a multimodal dataset designed for classifying six basic emotions. Additionally, HFM (Liu et al., 2022) is a multimodal dataset focused on recognizing high-level implicit emotion of sarcasm.

| Method | General | | | | | Knowledge | | | | | OCR & Chart | | | | | Vision-Centric | | | | |
|---|---|---|---|---|---|---|---|---|---|---|---|---|---|---|---|---|---|---|---|---|
| | Avg | $MME^P$ | MMB | $SEED^I$ | GQA | Avg | $SQA^I$ | $MMMU^V$ | $MathVista^M$ | AI2D | Avg | ChartQA | OCRBench | TextVQA | DocVQA | Avg | MMVP | RealworldQA | $CV\text{-}Bench^{2D}$ | $CV\text{-}Bench^{3D}$ |
| GPT-4V | 63.0 | 1409.4 | 75.8 | 69.1 | 36.8 | 65.2 | 75.7 | 56.8 | 49.9 | 78.2 | 77.4 | 78.5 | 64.5 | 78.0 | 88.4 | 62.4 | 50.0 | 61.4 | 64.3 | 73.8 |
| Gemini-1.0 Pro | - | 1496.6 | 73.6 | 70.7 | - | - | 79.5 | 47.9 | 45.2 | - | - | - | 65.9 | - | - | - | - | - | - | - |
| Gemini-1.5 Pro | - | - | - | - | - | - | - | 58.5 | 52.1 | 80.3 | - | 81.3 | - | 73.5 | 86.5 | - | - | 67.5 | - | - |
| Grok-1.5 | - | - | - | - | - | - | - | 53.6 | 52.8 | 88.3 | - | 76.1 | - | 78.1 | 85.6 | - | - | 68.7 | - | - |
| MM-1-8B | - | 1529.3 | 72.3 | 69.9 | - | - | 72.6 | 37.0 | 35.9 | - | - | - | - | - | - | - | - | - | - | - |
| MM-1-30B | - | 1637.6 | 75.1 | 72.1 | - | - | 81.0 | 44.7 | 39.4 | - | - | - | - | - | - | - | - | - | - | - |
| *Base LLM: Llama-3-Ins-8B* | | | | | | | | | | | | | | | | | | | | |
| Mini-Gemini-HD-8B | 72.7 | **1606.0** | 72.7 | 73.2 | 64.5 | 55.7 | 75.1 | 37.3 | 37.0 | 73.5 | 62.9 | 59.1 | 47.7 | 70.2 | 74.6 | 51.5 | 18.7 | 62.1 | 62.2 | 63.0 |
| LLaVA-NeXT-8B | 72.5 | 1603.7 | 72.1 | 72.7 | **65.2** | 55.6 | 72.8 | 41.7 | 36.3 | 71.6 | 63.9 | 69.5 | 49.0 | 64.6 | 72.6 | 56.6 | 38.7 | 60.1 | 62.2 | 65.3 |
| Cambrian-1-8B | **73.1** | 1547.1 | **75.9** | 74.7 | 64.6 | 61.3 | **80.4** | 42.7 | **49.0** | 73.0 | 71.3 | 73.3 | 62.4 | **71.7** | 77.8 | 65.0 | 51.3 | **64.2** | 72.3 | 72.0 |
| MODA-8B | 72.1 | 1535.9 | 73.8 | **74.9** | 63.0 | **61.5** | 80.4 | 43.1 | 48.8 | 73.6 | 72.0 | 74.3 | 65.2 | 70.4 | 78.1 | 66.0 | 52.6 | 64.1 | 73.5 | 73.8 |
| *Base LLM: Hermes2-Yi-34B* | | | | | | | | | | | | | | | | | | | | |
| Mini-Gemini-HD-34B | 76.2 | 1659.0 | 80.6 | 75.3 | 65.8 | 62.4 | 77.7 | 48.0 | 43.4 | 80.5 | 68.1 | 67.6 | 51.8 | 74.1 | 78.9 | 63.8 | 37.3 | 67.2 | 71.5 | 79.2 |
| LLaVA-NeXT-34B | 76.0 | 1633.2 | 79.3 | **75.9** | **67.1** | 62.5 | 81.8 | 46.7 | 46.5 | 74.9 | 67.7 | 68.7 | 54.5 | 69.5 | 78.1 | 64.0 | 47.3 | 61.0 | 73.0 | 74.8 |
| Cambrian-1-34B | **76.8** | **1689.3** | 81.4 | 75.3 | 65.8 | 67.0 | 85.6 | 49.7 | 53.2 | 79.7 | 71.9 | 75.6 | 60.0 | 76.7 | 75.5 | 68.5 | 52.7 | 67.8 | 74.0 | 79.7 |
| MODA-34B | 76.7 | 1639.2 | **82.3** | 75.8 | 66.2 | **69.5** | **88.1** | **52.5** | **54.0** | **83.4** | **74.7** | **79.8** | **62.7** | **78.3** | 78.2 | **69.9** | **53.8** | **68.5** | **75.8** | **81.3** |

*Table 2.* **Comparison of MODA with other leading MLLM framework on twelve perception benchmarks.** MODA outperforms other open-source models and achieves competitive performance on a number of benchmarks, compared to proprietary models such as GPT-4V, Gemini, and Grok-1.5. The reported numbers of leading MLLMs come from (Tong et al., 2024a).

## 4.2. Settings

We set the same experiment setting as (Tong et al., 2024a; Liu et al., 2023). We adopt CLIP (ViT-L/14) (Radford et al., 2021) as the visual encoder. For the foundational large language model, we choose models from different scales, *i.e.*, 8B: Llama-3-Instruct-8B (Dubey et al., 2024) and 34B: Hermes2-Yi-34B (Young et al., 2024). MODA is trained for 1 epoch with a batch size of 2048, using the AdamW (Loshchilov & Hutter, 2019) optimizer with a cosine learning rate schedule. The learning rate is set to 2e-5 for LLM and 2e-6 for visual encoder, respectively. The warmup rate is 0.03.

## 4.3. Ablation Study

To investigate the effectiveness of duplex attention alignment and modular attention mask, we conduct a component-wise ablation study in Table 1. For ablation studies, we train the MLLMs at the scale of 8B, with the base LLM of Llama-3-Ins-8B. For a fair comparison, all models are trained on 700K data samples for 1 epoch. We further discuss each component by conducting in-depth analyses of their variants to answer the following research questions.

- **RQ1:** How does the design of duplex attention alignment impact cross-modal feature transfer?

- **RQ2:** How does the modular attention mask address modality position bias and improve attention?

- **RQ3:** How do the proposed duplex attention alignment and modular attention mask respectively interact to

enhance multimodal attention?

- **Response to RQ1: Modality Axis Transfer** we analyze the effectiveness of duplex attention alignment in facilitating cross-modal feature transfer by examining its ability to align modality-specific features along a shared latent axis. This is motivated by the need to reduce modality gaps and ensure effective information exchange between modalities. We design experiments to test different variants of duplex attention alignment, such as using covariance matrices, attention head configurations, and linear vs. non-linear transformations.

- **Response to RQ2: Modality Position Bias** we investigate the role of the modular attention mask in addressing modality position bias and improving attention distribution. This analysis is crucial for understanding how the mask prevents attention collapse and ensures balanced contributions from all modalities. We experiment with different masking mechanisms, such as traditional infinity masking, fix-valued masking, and learnable masking. These variants are evaluated on tasks involving long sequences and imbalanced modality contributions, such as vision-centric perception and knowledge understanding.

- **Response to RQ3: Multimodal Attention Matrix** we analyze the interaction between duplex attention alignment and modular attention mask by studying their combined effect on the multimodal attention matrix. This is motivated by the hypothesis that the two components work synergistically to improve multimodal representation learning by enhancing both alignment and attention distribution. We de-

| Model | Cognition | | | | | | | | |
|---|---|---|---|---|---|---|---|---|---|
| Method | Avg | Instruction Adherence | Fluency | Coherence | Image-Text relevance | Response Accuracy | Personality Consistency | Knowledge Consistency | Tone Consistency |
| GPT-4 Turbo | 1.099 | 1.055 | 1.032 | 1.084 | 1.097 | 1.092 | 1.168 | 1.103 | 1.161 |
| Gemini 1.0 Pro | 1.021 | 0.999 | 1.007 | 1.028 | 1.009 | 1.013 | 1.052 | 1.013 | 1.050 |
| Claude 3 Opus | 1.157 | 1.127 | 1.070 | 1.149 | 1.167 | 1.146 | 1.219 | 1.168 | 1.213 |
| QWen-VL-Max | 1.028 | 1.014 | 1.012 | 1.035 | 1.034 | 1.029 | 1.042 | 1.021 | 1.041 |
| *Base: Llama-3-Ins-8B* | | | | | | | | | |
| Mini-Gemini-HD-8B | 0.878 | 0.884 | 0.942 | 0.898 | 0.864 | 0.853 | 0.855 | 0.876 | 0.852 |
| LLaVA-NeXT-8B | 0.968 | 0.971 | 0.988 | 0.980 | 0.966 | 0.967 | 0.966 | 0.964 | 0.939 |
| Cambrian-1-8B | 0.895 | 0.901 | 0.957 | 0.934 | 0.886 | 0.889 | 0.860 | 0.892 | 0.838 |
| MODA-8B | **0.972** | **0.976** | **0.992** | **0.985** | **0.970** | **0.972** | **0.970** | **0.969** | **0.945** |
| *Cognition-Specialized* | | | | | | | | | |
| MMRole-9B | 0.994 | 0.998 | 1.000 | 0.997 | **0.993** | 0.987 | 1.000 | **0.992** | **0.988** |
| MODA-8B | **0.995** | **1.000** | **1.001** | **0.999** | **0.993** | **0.989** | **1.001** | 0.991 | **0.988** |
| MMRole-9B (In-Test) | 0.999 | 1.000 | 1.000 | 0.999 | 0.997 | 0.989 | 1.012 | **0.997** | **0.997** |
| MODA-8B (In-Test) | **1.000** | **1.002** | **1.001** | **1.000** | **0.998** | **0.992** | **1.013** | 0.996 | 0.996 |
| MMRole-9B (Out-Test) | 0.981 | 0.992 | 0.999 | 0.993 | 0.979 | 0.981 | 0.963 | 0.977 | 0.962 |
| MODA-8B (Out-Test) | **0.984** | **0.995** | **1.002** | **0.996** | **0.981** | **0.983** | **0.970** | **0.980** | **0.965** |

*Table 3.* **Comparison of MODA with other leading MLLMs and cognition task-specialized methods on MMRole benchmark.** The numbers of leading MLLMs come from (Dai et al., 2025).

| Model | Emotion | | | | | | | | |
|---|---|---|---|---|---|---|---|---|---|
| Method | Avg | MVSA$^S$ (ACC) | MVSA$^S$ (F1) | MVSA$^M$ (ACC) | MVSA$^M$ (F1) | TumEmo (ACC) | TumEmo (F1) | HFM (ACC) | HFM (F1) |
| GPT-4V | 0.633 | 0.507 | 0.570 | 0.609 | 0.631 | 0.608 | 0.612 | 0.764 | 0.765 |
| Gemini 1.0 Pro | 0.646 | 0.634 | 0.637 | 0.699 | 0.657 | 0.598 | 0.582 | 0.674 | 0.683 |
| Claude 3 Opus | 0.628 | 0.626 | 0.613 | 0.635 | 0.629 | 0.580 | 0.574 | 0.679 | 0.687 |
| QWen-VL-Max | 0.643 | 0.647 | 0.645 | 0.669 | 0.627 | 0.565 | 0.595 | 0.696 | 0.701 |
| *Base: Llama-3-Ins-8B* | | | | | | | | | |
| Mini-Gemini-HD-8B | 0.482 | 0.423 | 0.571 | 0.487 | **0.643** | 0.246 | 0.395 | 0.498 | **0.593** |
| LLaVA-NeXT-8B | 0.576 | 0.591 | 0.593 | 0.617 | 0.607 | 0.547 | 0.533 | 0.572 | 0.551 |
| Cambrian-1-8B | 0.547 | 0.694 | 0.661 | 0.662 | 0.579 | 0.439 | 0.344 | 0.512 | 0.487 |
| MODA-8B | **0.588** | **0.702** | **0.705** | 0.628 | 0.619 | **0.559** | **0.548** | **0.585** | 0.563 |
| *Emotion-Specialized* | | | | | | | | | |
| M$^2$CL | - | 0.755 | 0.742 | 0.732 | 0.705 | 0.688 | 0.687 | - | - |
| MULSER | - | 0.757 | 0.755 | 0.739 | 0.738 | 0.775 | 0.775 | - | - |
| CMGCN | - | 0.733 | 0.720 | 0.697 | 0.683 | - | - | 0.875 | 0.841 |
| SPFVTE | - | 0.806 | 0.801 | 0.799 | 0.788 | - | - | 0.883 | 0.879 |
| MODA-8B | **0.841** | **0.810** | **0.803** | **0.802** | **0.790** | **0.778** | **0.778** | **0.885** | **0.881** |

*Table 4.* **Comparison of MODA with other leading MLLMs as well as emotion task-specialized methods on four emotion benchmarks.** The reported numbers of emotion-specialized methods come from their official manuscripts. The missed average performance of emotion-specialized methods due to missed datasets.

sign experiments that compare the joint use of these components against their individual use, as well as against baseline models without either component. Tasks such as question answering and multimodal summarization are chosen to simultaneously evaluate alignment and distribution.

### 4.4. Results

As shown in Table 2, Table 3, and Table 4, we demonstrate the main results on 21 popular benchmarks for multimodal perception, cognition, and emotion tasks, respectively.

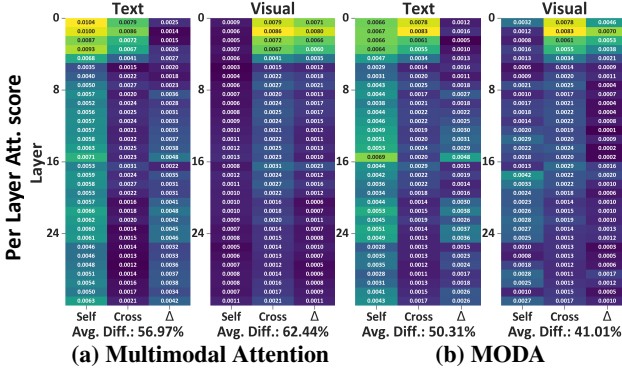

*Figure 4.* Analysis of multimodal attention and MODA. (a) Multimodal attention: average difference between self and cross-modal attention is 56.97% for text and 62.44% for visual. (b) MODA: average difference is 50.31% for text and 41.01% for visual.

● **Perception Benchmark.** To assess the effectiveness of our proposed model, we compare it against state-of-the-art Multimodal Large Language Models (MLLMs), including the Mini-Gemini-HD series, the LLaVA-NeXT series, and the Cambrian-1 series. We conduct a comparison under two settings, where we tune these MLLMs from 8B and 34B scale large foundation models. Our proposed MODA outperforms other models of similar scale, including LLaVA-NeXT and Cambrian, achieving an average improvement of 1.0 for the base Llama-3-Ins-8B model and 0.9 for the base Hermes2-Yi-34B model. In vision-centric and OCR tasks, which require fine-grained understanding, MODA consistently performs better, achieving a metric of 66.0 for the vision-centric average and 74.7 for OCR & Chart tasks. This highlights the model's ability in tasks demanding fine-grained perception, further reinforcing its superiority.

● **Cognition-specific & Emotion-specific Benchmark.** We evaluate a diverse set of MLLMs on both cognition-centric and emotion-centric benchmarks, designed to assess key dimensions of cognitive and emotional understanding across various aspects. On the cognition benchmark, MODA outperforms open-ended models of Cambrian-1 (0.981) and LLaVA-NeXT (0.979), achieving an average score of 0.996, and performs comparably to close-ended SOTA models like Claude 3 Opus (0.995). With cognition task-specific tuning, MODA achieves better performance, particularly excelling in fluency (0.999) and personality consistency (0.998). On emotion benchmarks, MODA outperforms open-ended models like Cambrian-1 (0.628) and LLaVA-NeXT (0.624), with an average score of 0.657, and achieves comparable performance to task-specialized SOTA models like SPFVTE (0.738) and MULSER (0.739). Notably, MODA excels in TumEmo (0.747) and HFM (0.753) benchmarks, demonstrating strong performance in emotion recognition tasks due to its ability to capture fine-grained emotional features and fine-grained details.

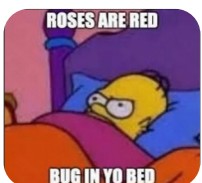

**Human:** Why is this image funny?

**Cambrian-1:** The humor in this image comes from the unexpected twist on the classic poem 'Roses are red, violets are blue.' Instead of a romantic or sweet sentiment, the poem is humorously altered to a more mundane and slightly unsettling scenario, which is a bug in the bed. This juxtaposition of a traditional romantic poem with a common, albeit slightly unpleasant, household occurrence creates a comedic effect.

**MODA:** The image is funny because it features a cartoon character, specifically Homer Simpson from the popular television show "The Simpsons," who is depicted in a humorous and unexpected situation. He is lying in bed with a red rose in his mouth, which is an unusual and comical scenario for the character. The image also includes a caption that reads "Roses are red, bug in yo bed," adding to the humor of the scene. This combination of elements creates a lighthearted and amusing visual that is likely to elicit laughter from viewers familiar with the character and the show.

**(a)** Fine-grained Perception

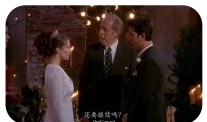

**Phoebe Buffay:** Look at this photo! Isn't it just like a fairy-tale wedding, until, you know, it wasn't?

**Ross Geller played by Cambrian-1:** Phoebe, you're right, the wedding does look like a fairy-tale scene, but it's not quite perfect. You see, the bride, Emily, is mistakenly called Rachel by the groom, Ross. It's a bit of a mix-up, but it's all good. It's just one of those little hiccups that happens in real life, you know?

**Ross Geller played by MODA:** Oh, Phoebe, you're right. It's like a fairy-tale wedding, with beautiful bride and handsome groom, and the elegant setting. But, as we all know, fairy tales don't always have a happy ending. In this case, Ross made a mistake that changed the course of wedding, and it's a moment that will be remembered by everyone. It's a reminder that even in the most perfect of settings, life can throw us curveballs that we have to navigate.

**(b)** Cognitive Conversation Analysis

**Post:** And just like that the weekends gone.

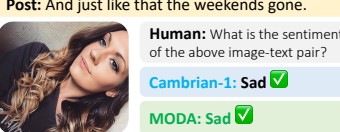

**Human:** What is the sentiment of the above image-text pair?

**Cambrian-1: Sad** ✅

**MODA: Sad** ✅

**Post:** My photogenic little man

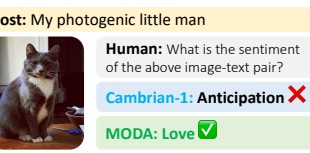

**Human:** What is the sentiment of the above image-text pair?

**Cambrian-1: Anticipation** ❌

**MODA: Love** ✅

**Post:** New GIPHY!

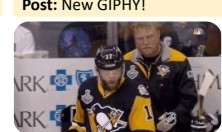

**Human:** What is the sentiment of the above image-text pair?

**Cambrian-1: Happy** ❌

**MODA: Angry** ✅

**(c)** Emotion Understanding

*Figure 5.* Visualization results of state-of-the-art (SOTA) Multimodal Large Language Models (MLLMs) and our proposed MODA on fine-grained multimodal tasks, including perception, cognition, and emotion understanding.

● **Analysis of Attention.** We analyze the attention distribution to evaluate the effectiveness of the proposed MODA in terms of attention distribution (Fig. 4). We conduct experiments on fine-grained understanding tasks, *i.e.*, emotion recognition based on visual-textual affective cues. Baseline attention exhibits imbalanced scores, particularly in deeper layers, highlighting its tendency to over-focus on dominant modalities while neglecting others (*i.e.*, visual affective cues). Our MODAachieves consistently high across all layers, preventing degradation and ensuring balanced contributions from all tokens and modalities. This improvement directly correlates with superior performance on multimodal tasks, particularly those requiring fine-grained understanding, such as TumEmo and OCRBench, demonstrating MODA's ability to avoid attention collapse and capture multimodal interactions effectively.

● **Visualization** To highlight the advantages of Modular Duplex Attention (MODA), we design a visualization experiment focusing on output answers, showcasing its capacity to generate fine-grained and accurate responses. The output answers (Figure 5) further demonstrate MODA's superior capacity for fine-grained understanding. By accurately capturing the humorous twist on the classic poem, MODA has demonstrated its unparalleled ability to reason about fine-grained multimodal details. These results underscore MODA's transformative potential in advancing the state-of-the-art in multimodal understanding, where it seamlessly integrates information across modalities to achieve exceptional precision in complex cognitive and emotional tasks.

**Johnny:** A month ago he bought the rights to this book, a best seller. The main character is a guy just like me. I wouldn't even have to act, just be myself. Oh Godfather, I don't know what to do, I don't know what to do [crying]

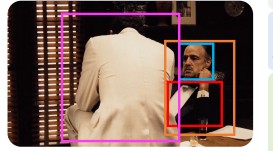

**Human:** How is the Godfather looks like?

**Cambrian-1:** The Godfather, as depicted in the image, is portrayed by an actor in a classic black tuxedo, which is a traditional and iconic look for the character.

**MODA:** The Godfather is dressed in a classic black tuxedo, which is emblematic of his role as a powerful and respected figure in the film. His expression is one of concern, as he listens to Johnny's dilemma, indicating his deep involvement in Johnny's life and the weight of his responsibilities.

**Human:** What does this scene reveal about the Godfather's influence over people in his inner circle?

**Cambrian-1:** This scene from 'The Godfather' illustrates the profound impact the character, Don Corleone, has on those around him. It shows how his influence extends beyond his immediate family, affecting even those who are not directly related to him.

**MODA:** This scene reveals the Godfather's influence over people in his inner circle as a significant overwhelming force. The Godfather's presence in the room, along with his stern demeanor, further emphasizes his control over the situation and the characters around him. The scene captures the tension and emotional turmoil that can arise from being under the Godfather's influence, highlighting the power dynamics at play within the narrative.

(a) Understanding *Godfather*

**Human:** How might the Godfather act to proceed with the next conversation? Give me a few possible options.

**Cambrian-1:** Given the context of the image, which suggests a scene from "The Godfather," the character portrayed by the man in the foreground might proceed with the **next conversation** by either **continuing to listen** intently to the other character, who seems to be speaking, or by **interjecting with a question or comment** to steer the conversation in a particular direction.

**MODA:** The Godfather, upon witnessing Johnny's distress, might respond with a **combination of empathy and strategic thinking.** He could offer a comforting word, such as, "It's a lot to take in, but we'll figure it out together." Alternatively, he might shift the conversation to a more **practical matter**, like **discussing the logistics of the book's adaptation or the potential benefits of the project.** Another possibility is that he might offer **advice on how to handle the situation**, such as seeking legal counsel or consulting with other members of the family. The Godfather's response would likely be measured and thoughtful, reflecting his role as a leader.

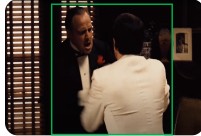 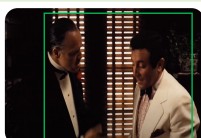 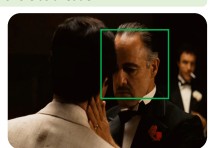

(b) Planning for *Godfather*

*Figure 6.* MODA-enabled apps in *The Godfather*. (a) With a deep and fine-grained understanding of conversation, MODA captures both the emotional and cognitive states of the character. (b) MODA further simulates the *Godfather*'s strategic thinking, planning the next steps while considering the character's traits.

# 5. Discussion

MODA-based MLLM can boost various downstream directions. Here, we envision two potential uses.

• *Godfather*-centric Understanding. Leveraging the fine-grained understanding of multimodal content, MODA demonstrates enhanced comprehension capabilities that facilitate human-centric interpretation. Integrated with the strong sensory grounding capacity, MODA can effectively process and interpret high-level human cognition and emotion. Notably, we present the first demonstration of a Multimodal Large Language Model (MLLM) that captures emotions expressed by individuals, going beyond simple classification into discrete categories. As illustrated in Fig.6 (a), MODA successfully captures the micro-expressions of the *Godfather*, reflecting the character's personality and the cultural context embedded within the scene.

• Conversation with *Godfather*. Further, the comprehensive understanding enabled by MODA enhances its capacity to plan conversations. We integrate MODA into a conversation system between the human and the agent in three stages: description, analysis, and planning. As shown in Fig.6 (b), MODA takes the historical conversation context (e.g., video keyframe, transcript, and audio) as input and generates the desired target (e.g., behavior, characteristics, and emotion). This allows the model to simulate a conversation flow, as in the *Godfather* scenario, where the agent responds with strategic plan based on the established emotional context.

# 6. Conclusion

This work introduces the MOdular Duplex Attention (MODA) to tackle attention deficit disorder in multimodal large language models, characterized by inconsistent cross-modal attention and layer decay. MODA enhances multimodal perception, cognition, and emotion understanding by modularly processing diverse data streams, outperforming existing MLLMs across 21 benchmark datasets. This advancement not only improves modality alignment but also supports deeper cognitive and emotional insights, with source code and demo available for further exploration.

# Acknowledgement

This work was supported by the National Natural Science Foundation of China (No. 623B2056), the Natural Science Foundation of Tianjin, China (No.24JCZXJC00040), the Fundamental Research Funds for the Central Universities, the Supercomputing Center of Nankai University (NKSC). We sincerely thank the reviewer team (cYUZ, XinM, Mf8z, and Ack2) for their invaluable feedback to improve our manuscript.

# Impact Statement

This paper introduces a novel multimodal attention mechanism designed to enhance Multimodal LLMs for fine-grained content understanding. However, as with most MLLMs, the quality of MODA's output is influenced by the fine-tuning data and the quality of the base models, which may result in the generation of low-quality or hallucinated content. Such outputs could potentially be harmful, and users are advised to interpret the results with caution, adhering to licensing restrictions, with commercial use explicitly prohibited. All the personal information is anonymized or obfuscated to ensure confidentiality.

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
