# OpenReview forum: "MODA: MOdular Duplex Attention for Multimodal Perception, Cognition, and Emotion Understanding"
_ICML.cc/2025/Conference — ICML 2025 spotlightposter_

### Official Review · Reviewer_cYUZ · 2025-03-11

**Overall Recommendation:** 5

**Summary:**

The paper identifies the attention deficit disorder problem in SOTA MLLMs, characterized by inconsistent cross-modal attention and layer-by-layer decay of attention activation. Then, the authors introduce a linear-based attention mechanism that simultaneously conducts inner-modal refinement and inter-modal interaction. Experimental results show an improvement on fine-grianed detail understanding. The evaluation setting follows Cambrian-1, MMRole, and some emotion benchmarks.

**Claims And Evidence:**

-The claimed linear complexity is not supported by any details. The paper does not provide detailed analysis of MODA's computational cost, especially compared to simpler baselines.

-While the modular masked attention and duplex alignment are well-motivated, the paper does not explore alternative designs or compare MODA to other attention mechanisms (e.g., sparse attention or low-rank attention).

**Essential References Not Discussed:**

N/A

**Experimental Designs Or Analyses:**

-The paper does not provide detailed analysis of MODA's computational cost (e.g., FLOPs, memory usage, or inference time) compared to simpler baselines.

**Methods And Evaluation Criteria:**

-Yes, the evaluabtion setting follows The evaluation setting follows Cambrian-1, MMRole, and some emotion benchmarks.

**Other Comments Or Suggestions:**

N/A

**Other Strengths And Weaknesses:**

see above

**Questions For Authors:**

see above

**Relation To Broader Scientific Literature:**

-The paper demonstrates MODA's effectiveness across a wide range of tasks, including perception, cognition, and emotion understanding, setting new benchmarks in multimodal understanding.

-The paper highlights the potential of MODA to advance multimodal understanding in real-world applications, such as human-computer interaction, robotics, and healthcare.

-Identify the Attention Deficit Disorder Problem for multimodal large language model area.

**Theoretical Claims:**

-The computation method of attention distribution is not mentioned with details

---

> ### Author Rebuttal · Authors · 2025-04-01
>
> ## **Response to Reviewer Ack2**
>
> Thank you for your insightful comments and questions. For your reference, we summarized the main results and included the attached file in our response to Reviewer XinM.
>
> **A1. Analysis for linear complexity of duplex attention alignment**
>
> **(1) Complexity Analysis:** As suggested, we provide the complexity analysis of the proposed duplex attention alignment. Given a multimodal token sequence $X\in\mathbb{R}^{N\times D}$, the computation cost comes from three operations, including Gram Matrix and alignment. In the Gram matrix, the computation cost is $O(D^2N)$. In the alignment, the computation cost is $O(DN)$, which in total is $O(D^2N+DN)=O(D^2N)$. Therefore, the proposed duplex attention alignment has a linear complexity in terms of token length, thus yielding a length extrapolation ability towards long multimodal context.
>
> **(2) Complexity comparison with other attention mechanism:** In contrast, the baseline attention owns a nature of $O(DN^2)$ due to the similarity computation between each pair of tokens. Therefore, they have a higher computation complexity of $O(DN^2)$, where in most cases $N>>D$, especially for multimodal tokens, the number can be up to 128000 (e.g., max model len of  Qwen2.5-VL).
>
> **A2. Ablation study on Attention mechanism**
>
> (1) As suggested, we further conduct a comparison between our MODA and other attention mechanisms, including baseline attention and SOTA attention mechanism. As shown below, MODA shows better performance among leading attention mechanisms due to its balance on multimodal tokens, while sparse and baseline attention may wrongly focus on the textual part.
>
> | Model                      |  G   |  K   |  O   |  V   |
> | -------------------------- | :--: | :--: | :--: | :--: |
> | MODA                       | 69.3 | 48.3 | 67.0 | 54.3 |
> | DeepSpeed Sparse Attention | 67.8 | 47.6 | 63.3 | 48.1 |
> | Multi-head Attention       | 63.6 | 44.0 | 60.8 | 38.0 |
>
> (2) Besides, as shown in Tab.1 of the manuscript, we provided an ablation study on alternative designs from three aspects: how to align attention tokens (Tab.1b), how to fuse attention tokens (Tab.1c), and how to mask attention  (Tab.1d).
>
> **A3. Computation cost:** As suggested, we compare the computation cost of MODA and other attention mechanisms in terms of FLOPs, memory, and latency as below. We can observe that MODA achieves better performance on fine-grained understanding tasks (e.g., four vision-centric benchmarks) with a few increased computation costs.
>
> | Model      | FLOPs  | MACs  | Latency | Vision-Centric |
> | ---------- | :----: | :---: | :-----: | :------------: |
> | MODA       | 134.9T | 76.7T |  2.04s  |      66.0      |
> | LLaVA-NeXT | 123.3T | 61.6T |  1.87s  |      56.6      |

---

> > ### Comment · Reviewer_cYUZ · 2025-04-02
> >
> > Thanks to the response from the authors. After reading other reviews and responses, my concerns have been resolved. I am happy to see the significant margin over the latest baseline, benefiting from the novel insights on attention mechanisms. To the best of my current knowledge, the key insight might present positive impacts in the VLM area. Therefore, I lean towards a stronger acceptance and will upgrade my rating accordingly.

---

> > > ### Author Response · Authors · 2025-04-02
> > >
> > > Dear reviewer cYUZ,
> > >
> > > Thank you for kindly recognizing our contributions to VLM and providing invaluable comments on this work. We authors greatly appreciate the efforts you have made to improve our manuscript. If accepted, we will include `cYUZ` in our acknowledgments.
> > >
> > > Best Regards, Authors of paper 10155

---

### Official Review · Reviewer_ACk2 · 2025-03-12

**Overall Recommendation:** 3

**Summary:**

This paper proposes a MOdular Duplex Attention (MODA) for multimodal perception, cognition and emotion understanding. The proposed method is evaluated and the paper is well organized.

**Claims And Evidence:**

yes

**Essential References Not Discussed:**

no

**Experimental Designs Or Analyses:**

yes

**Methods And Evaluation Criteria:**

yes

**Other Comments Or Suggestions:**

no

**Other Strengths And Weaknesses:**

1. Introduction
 (1) Although the authors propose the phenomenon of deficit disorder attention, the discussion of its underlying causes and theoretical foundations is weak. The article focuses more on the description of the phenomenon and lacks strict mathematical definitions and theoretical explanations, which makes the concept appear to be more of an analogy than a well-proven theoretical basis.
(2) only mentions the inadequacy of MLLM for advanced tasks, but does not provide a detailed compendium and comparison of current research methods and advances to address this inadequacy.

2. Related work
When presenting related work, it is mostly descriptive, without clearly stating the direct correlation and difference between these studies and the methods proposed in this paper, and fails to clearly demonstrate the unique advantages of MODA in dealing with the problems of inconsistent attention and layer-by-layer attenuation.

3. Methods
(1) For the specific implementation of V-Aligner and T-Aligner, only the mapping based on the basis vectors of modal space is mentioned, but the specific form of the mapping function and why the basis vectors are used are not explained in detail. There is a lack of detailed description of the fusion method of the mapped tokens, such as whether weighted summing and splicing are used, as well as the parameter estimation and optimization strategy in the fusion process.
(2) Insufficient explanation for some formulas. For example, “||” in Equation 5.
(3) Although the attention is divided into two parts, self-modal and cross-modal, how to coordinate these two parts in the overall architecture and how to deal with their interactions and information fusion are not described clearly enough. There is a lack of detailed discussion on how the modules in the overall process work together.
(4) There is a lack of specific guidance and examples on how to integrate MODA into existing multimodal large-scale language models, which cannot provide effective references for actual model development and application.

4. Experimental Part
(1) The description of the experimental setup is not exhaustive enough, and key settings such as data preprocessing, hyperparameter selection, and training details (e.g., hardware environment, running time) are not fully developed.
(2) Despite the ablation experiments, there is a lack of detailed discussion on the individual contributions and interactions of the two modules (Duplex Attention Alignment and Modular Attention Mask) to the overall performance.
(3) In the comparison with existing SOTA models, only a large amount of performance data is given, while the lack of analysis of the advantages and disadvantages of each model on different tasks makes the discussion of the reasons for the performance improvement insufficient.
(4) The experimental results present more data, but fail to discuss in depth the performance differences between different tasks and metrics, potential bottlenecks, and directions for future improvement, limiting a comprehensive understanding of the advantages and limitations of the method.

**Questions For Authors:**

no

**Relation To Broader Scientific Literature:**

The authors propose a novel modular and duplex attention mechanism which could be used in multimodal LLMs.

**Theoretical Claims:**

yes

---

> ### Author Rebuttal · Authors · 2025-04-01
>
> ## **Response to Reviewer Ack2**
>
> Thank you for your insightful comments and questions. For your reference, we summarized the main results and included the attached file in our response to Reviewer XinM.
>
> **A1&A2. Discussion on DDA**
>
> (1) **Actually, DDA can be interpreted from the perspective of a multimodal token graph.** The core of DDA, the attention mechanism, builds the linkage across multimodal tokens by computing the similarity of each pair of tokens [A,B]. The attention output is yielded by a weighted sum among tokens. Therefore, the computation can be seen as a densely connected direct graph through layer-by-layer attention, and the DDA problem can be reformulated as a mislinked graph problem, where the multimodal token flows to the textual part and misses the visual token from the input layer. DDA, therefore, decreases the interaction between visual and textual modality, and throws away 17% of important visual cues and is lost in the last embedding layer of MLLM. For the detailed diagram of multimodal token flow, please refer to Fig.1 in the attached file.
>
> (2) **Formula definition of DDA.** Given the visual tokens $x_v^l$ and text tokens $x_t^l$ in the block $l$, the multimodal attention builds the link from two parts (*i.e.*, self-modal $x_t^l \rightarrow x_t^{l+1}, x_v^l \rightarrow x_v^{l+1}$ and cross-modal $x_t^l \rightarrow x_v^{l+1}, x_v^l \rightarrow x_t^{l+1}$ ),  where the links are commonly implemented by the pair-wise token similarity and weighted sum. However, the modality gap between tokens decrease the magnitude of links, as we observed, the link value of $x_v^l \rightarrow x_v^{l+1}$ and $x_t^l \rightarrow x_v^{l+1}$ decays exponentially with depth ($\alpha_{v \rightarrow v}^l \propto \gamma^l,\gamma \neq 1$). This misalignment propagates layer-wise, causing the cumulative error in cross-modal interaction to grow as $\mathbb{E}_L = \sum_l \gamma^l \epsilon_l$, where $\epsilon_l$ denotes the layer-specific alignment error. This phenomenon aligns with theoretical insights in [B], where pure attention mechanisms suffer from **rank collapse** – a critical factor exacerbating the attention distribution.
>
> (3) **MODA vs existing methods:** To address this issue, we propose MODA, which introduces modality-aware gating and layer-wise error compensation to mitigate DDA. Unlike existing methods, MODA dynamically adjusts cross-modal interactions through modality-specific gates and compensates for alignment errors using adaptive propagation mechanisms. Further, MODA extracts the modality-specific feature and introduces the gram matrix as basis vectors for adaptive mapping. Thanks for the suggestions and we will re-organize the literature review in the introduction and related work.
>
> [A] On the role of attention masks and layernorm in Transformers, NeurIPS, 2024
>
> [B] Attention is not all you need: Pure attention loses rank doubly exponentially with depth, ICML, 2021
>
> **A3-1. Mapping function in V&T-aligner:** The mapping functions in V-Aligner and T-Aligner are designed to address modality alignment issues and exploit four alternative designs as shown in Tab.1c of our manuscript. We explore direct replacement ($X_a$ as the original feature, $X_p$ as the mapped feature), concatenation, and element-wise addition for token fusion. The use of basis vectors enables structured modality space representation, improving alignment.
>
> **A3-2. Formulas:** $||$ in Equ.5 represents the normalization operation.
>
> **A3-3. Self-attention&cross-attention:** We modularize the influence of self- and cross-attention separately from two aspects: pull the tokens by alignment and correct the focus by masking.
>
> **A3-4. MODA for broader MLLMs:** MODA can be integrated into the existing MLLMs by simply replacing each attention layer in the Transformer block. In our paper, we implement the most commonly used pre-norm type MLLM and verify its effectiveness on LLaMA-based MLLM. As a result, MODA can be integrated into the LLaVA series and Vicuna-based, Yi-based, Wizardlm2-based MLLMs because they share the same architecture as the implemented one of MODA. For clarity, we have prepared a torch-style pseudocode in Alg. 2 of the attached file.
>
> **A4-1. Detailed experimental setup:** As suggested, we provide all the experimental settings, including the model, data, hyperparameter, and details. Due to the character limits, please refer to the Q1 of reviewer XinM.
>
> **A4-2&A4-3&A4-4. Analysis and Discussion of MODA:** We thank the reviewer for the feedback. To address this, we added a detailed analysis of the advantages and disadvantages of each model and module across perception, cognition, and emotion tasks. MODA, as most MLLMs are, suffer from the risk of generating counterfactual responses, leading to false narratives or misrepresentations.

---

### Official Review · Reviewer_XinM · 2025-03-13

**Overall Recommendation:** 5

**Summary:**

This paper proposes a novel attention mechanism called Modular Duplex Attention (MODA) to tackle the attention inconsistency problem in Multimodal Large Language Models (MLLMs). MODA showcases outstanding performance in multimodal perception, cognition, and emotion understanding tasks. Specifically, the 34B version of MODA outperforms GPT-4V comprehensively.

**Claims And Evidence:**

Yes, the claims made in the submission are supported by clear and convincing evidence.

**Essential References Not Discussed:**

none

**Experimental Designs Or Analyses:**

Yes, I have checked the soundness and validity of the experimental designs and analyses in the paper. In the ablation study, different components of MODA, such as duplex attention alignment and modular attention mask, were systematically removed or modified. This design is valid as it helps to isolate the impact of each component on the overall performance, answering important research questions about their individual contributions.

**Methods And Evaluation Criteria:**

Yes, the proposed methods and evaluation criteria make eminent sense for the problem and application at hand.

**Other Comments Or Suggestions:**

none

**Other Strengths And Weaknesses:**

Weakness:
1. As mentioned in **Supplementary Material**, I wanted to know if the model only trained on a SFT stage or followed a two stage training process like common models. Also, I'd like to know more details like learnable parameters during the training process.

**Questions For Authors:**

I was wondering why MODA has such a significant improvement in emotion tasks. I'm willing to raise my score if deeper explanations and training details can be provided.

**Relation To Broader Scientific Literature:**

Prior studies have highlighted the importance of attention mechanisms in multimodal learning, but faced issues with inconsistent cross-modal attention. MODA builds on these by specifically addressing this problem.

**Theoretical Claims:**

Yes, I've checked the theoretical claims' proofs. For MODA, the duplex attention alignment's math based on gram matrix vectors is logical. The modular attention mask's equations and strategies are well - defined. The use of the normed gram matrix in both components is sound. Overall, the proofs are correct and support the MODA mechanism.

---

> ### Author Rebuttal · Authors · 2025-04-01
>
> ## **General Response**
>
> We sincerely appreciate all the Reviewers and the Area Chair for their time and effort in reviewing our paper. Following the valuable suggestions and insights provided in the reviews, we summarize **the additional results and evidence** included in the rebuttal based on the reviewers' suggestions:
>
> - We provided the theoretical explanation of the deficit disorder attention problem and illustrated the diagram **[A1&A2 for Ack2 and Fig.1 of the attached file]**.
> - We conducted experiments on 9 benchmark datasets to discuss the negative impact of the critical issue of MLLM, i.e., the deficit disorder attention problem **[A1 for Mf8z and Fig.2 of the attached file]**.
> - We conducted more qualitative comparisons with SOTA MLLMs **[Fig.3 of the attached file]**.
> - We provided the pseudo-code for attention score computation and MODA for clarity **[A6 for Mf8z, A3-4 for Ack2,  and Alg. 1 & Alg. 2 of the attached file]**.
> - We provided the computational complexity analysis of the proposed module and conducted comparisons among existing MLLMs and alternative attention mechanisms **[A1 & A2 & A3 for cYUZ]**.
> - We provided the detailed experimental setup and illustrated more information, including training time and hardware **[A1 for XinM]**.
> - We corrected the mentioned typos and carefully checked our manuscript again. Moreover, we also modified the figures and added the discussions as suggested **[Fig.4 of the attached file]**.
>
> Additionally, we follow the guideline of ICML 2025 and attach the figures as well as the pseudo-code in the [`anonymous link`](https://anonymous.4open.science/r/icml25_rebuttal-F06A/README.md): https://anonymous.4open.science/r/icml25_rebuttal-F06A/README.md
>
> ---
>
> ## **Response to Reviewer XinM**
>
> Thank you for your insightful comments and questions.
>
> **A1. Training details, including backbone, data, hyperparameter, and hardware&time**
>
> (1) **Same setting as Cambrian-1:** To ensure a fair comparison, we adopt the same experiment setup, including the backbone and data, as Cambrian-1. Specifically, our training begins with a one-stage SFT following pretraining, which utilizes the pre-trained ViT and adaptor. During the training, we unfreeze the LLM backbone to fine-tune the attention part in the LLM and fully exploit the power of MODA.
>
> (2) **Details.** We attach a configuration table below, which outlines the backbone, data, hyperparameter, and details during the training process. We will include details about training in the revised manuscript.
>
> |               | Backbone        |                          | Data        | Param. |        |        | Details      |          |
> | ------------- | --------------- | ------------------------ | ----------- | ------ | ------ | ------ | ------------ | -------- |
> | **Model**     | **LLM**         | **Vision**               | -           | **lr** | **wd** | **bs** | **Hardware** | **Time** |
> | MODA-8B       | LLaMA3-Ins-8B   | OpenAI CLIP ViT-L/14@336 | Cambrian-7M | 2e-5   | 0      | 1024   | 2x 8 A800    | 6 days   |
> | Cambrian-1-8B | LLaMA3-Ins-8B   | 4 Vision Encoders*       | Cambrian-7M | 2e-5   | 0      | 512    | 128 TPUv4    | -        |
> | MODA-34B      | Hermes-2-Yi-34B | OpenAI CLIP ViT-L/14@336 | Cambrian-7M | 2e-5   | 0      | 2048   | 4x 8 A800    | 14 days  |
> | Cambrian-34B  | Hermes-2-Yi-34B | 4 Vision Encoders*       | Cambrian-7M | 2e-5   | 0      | 1024   | 512 TPUv4    | -        |
>
> - Backbone: Cambrian-1 uses 4 vision encoders* including OpenAI CLIP ViT-L/14@336, SigLIP ViT-SO400M/14@384, DINOv2 ViT-L/14@518, and Open-CLIP ConvNeXt-XXL@1024. In contrast, we only use OpenAI CLIP ViT-L/14@336 as other popular MLLMs.
> - Data: We follow the same setting as Cambrian and train our MODA by Cambrian-7M.
> - Hyperparameter: We follow the common setting and train the model by using different lr for LLM and vision encoder.
> - Hardware&time: For the 8B model, we use 2x A800 nodes to train for 6 days. For the 34B model, we use 4x A800 nodes to train for 14 days.

---

> > ### Comment · Reviewer_XinM · 2025-04-08
> >
> > The responses has addressed my concerns, and I’m willing to raise my score.

---

> > > ### Author Response · Authors · 2025-04-08
> > >
> > > Dear reviewer XinM,
> > >
> > > Thank you for kindly providing invaluable suggestions on this work. We authors greatly appreciate the efforts you have made to improve our manuscript. We will add the training details to the revised manuscript. If accepted, we will include `XinM` in our acknowledgments.
> > >
> > > Best Regards, Authors of paper 10155

---

### Official Review · Reviewer_Mf8z · 2025-03-17

**Overall Recommendation:** 3

**Summary:**

The paper identifies a critical limitation in MLLMs, where inconsistent attention across layers leads to errors in fine-grained emotion understanding ("deficit disorder attention problem"). To address this, the authors propose MOdular Duplex Attention (MODA), which separates attention into self-modal and cross-modal components, each governed by a dedicated modulated attention mask. Extensive evaluations on various benchmark datasets demonstrate MODA’s effectiveness.

## update after rebuttal

Thank you to the authors for their responses. Most of my concerns have been addressed, and I am happy to raise my rate.

**Claims And Evidence:**

- The authors introduce the deficit disorder attention problem, arguing that it leads to neglect fine-grained details. However, this claim seems to be an overgeneralization.
- Also, the visualizations of attention scores across different modalities (Figures 1(c), 2(c), and 5) may not provide convincing evidence of an inherent problem. Different modalities inherently contain varying levels of similarity between tokens. For instance, in visual inputs, adjacent tokens often exhibit higher similarity due to the spatial structure of images, naturally resulting in smaller self-attention scores. This phenomenon may not necessarily indicate a limitation but rather a characteristic of multimodal attention behavior.

**Essential References Not Discussed:**

No major references appear to be missing.

**Experimental Designs Or Analyses:**

Some details are missing from the experimental results section.
- How do the authors compute the attention scores? Are these derived from a single sample or aggregated from multiple samples? Additionally, which dataset(s) were used for this analysis?
- What does [M] represent in Table 1?
- In Line 184, the term "the original ones" is vague.
- While the paper states that the mask is split into $\mathbf{M}^m$ and $\mathbf{M}^\bar{m}$, it is unclear what the mask itself represents.

**Methods And Evaluation Criteria:**

The authors claim that their proposed approach enhances fine-grained content understanding and support this with qualitative results (e.g. Figure 6). However, they do not provide a direct comparison with other MLLMs.

**Other Comments Or Suggestions:**

It might be useful to provide a more explanation of the Gram matrix, as MODA relies on it.

**Other Strengths And Weaknesses:**

**Weaknesses:**
- The paper requires proofreading as there are several missing details (as noted in my comments above), and some sentences are incomplete or unclear.
```(Line 204) To alleviate the collapsed attention matrix and prevent it from under-smoothed tokens. We propose a modular attention mask that chooses to store unnecessary attention values on these pseudo-attention scores.```

**Questions For Authors:**

Please check my comments above.

**Relation To Broader Scientific Literature:**

It may be benefitial for multimodal perception, cognition, and emotion understanding.

**Theoretical Claims:**

The primary concern from this reviewer is in Section 3, particularly regarding the clarity and alignment of the model description.
- The overall approach and flow of the proposed model architecture is unclear. While Figure 3 is intended to illustrate the framework, it does not clearly align with the explanation in Section 3, making it difficult to understand how the components interact.
- In Figure 2(a), what do the x-axis and y-axis represent? How does the visualization support their claim about attention inconsistency across layers (Lines 133–137)?

---

> ### Author Rebuttal · Authors · 2025-04-01
>
> ## **Response to Reviewer Mf8z**
>
> Thank you for your insightful comments and questions. For your reference, we summarized the main results and included the attached file in our response to Reviewer XinM.
>
> **A1. Claims on DDA**
>
> (1) Actually, DDA is supported by evidence from two aspects: rationale and observation.
>
> - **Rationale**: Based on the graph theory, the interaction of multimodal tokens are discorrupted from the input level, where most important visual tokens are throw away. Further, the layer-by-layer propagation introduces the coefficient of accumulated ignorance. The illustration is included in Fig.1 of the attached file.
> - **Observation**: First, attention scores exhibit a significant bias toward the textual modality, as shown in Fig.2a, where visual features are underrepresented. Second, Fig.2b&2c highlight a clear layer-wise attention decay, with attention inconsistencies becoming more pronounced in deeper layers. Finally, qualitative analyses in Fig.4&6 reveal that baseline models fail to capture subtle multimodal cues, resulting in incorrect or overly generic responses.
>
> (2) We thank the reviewer for the valuable comments and will **clarify this** in the revised manuscript accordingly.
>
> **A2. Discussion on DDA**
>
> (1)  **Similar visual tokens lead to high attention scores.** The authors would like to clarify that the statement of the reviewer 'adjacent tokens often exhibit higher similarity due to the spatial structure of images, naturally resulting in smaller self-attention scores.' is *wrong*. In fact, the similar visual tokens have a high attention score [A], according to the definition of attention $A = Softmax(QK/\sqrt{d})\propto QK$. $QK$ represents the similarity between tokens and is directly proportional to the attention score. Therefore, DDA suffers from imbalanced attention scores. The visual part is assigned a low attention score, leading to the neglect of important visual details.
>
> (2) **DDA brings negative impact in neglecting fine-grained details.**  We use a grounded MLLM (SA2VA) prompt by the ground truth answer to segment the key regions in visual content. Experimental results on 9 vision-centric perception, cognition, and emotion benchmarks revealed that the SOTA MLLM throws away 17% of the crucial visual tokens. For visualization results, please refer to Fig.2 of the attached file.
>
> (3) **DDA limits real-world applications.**  The imbalanced multimodal attention introduces bias in the flow of tokens, which can lead to inadequate token fusion and yield hallucination results [B]. As shown in Fig.1a &1b of our original manuscript, the model failed to recognize the waiter in the image background, posing critical challenges for practical scenarios, including OCR and conversational agent tasks.
>
> [A] Attention is all you need, NIPS, 2017.
>
> [B] Mitigating modality prior-induced hallucinations in multimodal large language models via deciphering attention causality, ICLR, 2025
>
> **A3. Comparison on fine-grained understanding:** We would like to clarify that our manuscript already *INCLUDES BOTH quantitative and qualitative direct comparisons with other leading MLLMs* in fine-grained content understanding, as presented in Tab.2, 3, 4, and Fig.4, 7.
>
> (1) **Tab.2**: We compare MODA with 6 close-sourced and 6 open-sourced MLLMs (GPT4V, Gemini 1.5 Pro, Grok-1.5, MM-1-30B, Mini-Gemini-HD, LLaVA-NeXT, Cambrian-1) on vision-centric and OCR tasks, which rely on fine-grained cue understanding.
>
> (2) **Tab.3&4**: We compare MODA with 4 close-sourced and 7 open-sourced MLLMs (GPT4o, Gemini 1.5 Pro, Claude 3 Opus, Owen-VL-Max, Mini-Gemini-HD, LLaVA-NeXT, Cambrian-1, MMRole)  on cognition and emotion tasks requiring contextual understanding.
>
> (3) **Fig.4&7**: we provide qualitative comparisons with SOTA Cambrian-1.
>
> (4) **Fig.3 of the attached file:** Additionally, we conduct further quantitative comparisons with Cambrian-1 on human-centric understanding and planning tasks.
>
> **A4. Framework and pipeline in Fig.3 and Sec. 3:** Thanks for the reminder and we include the modified pipeline and caption as Fig.4 of the attached file. Besides, we will add an overview section in Sec 3.3.
>
> **A5. Illustration of Fig.2:** The x-axis and y-axis represent the attention values and the corresponding probability. The x-axis ranges from 1e-4 to 1e-1 and is a logarithmic scale.
>
> **A6. Attention score:** Following [C], we compute token-level attention scores on all the testing set and perform average. The pseudocode is included as Alg.1 in the attached file.
>
> [C] Efficient streaming language models with attention sinks, NeurIPS, 2024
>
> **A7&A8. Writing:** (1) [M] replaces modular masked attention with learnable tokens to control attention distribution. (2) The term refers to the residual features before Gram matrix mapping. (3) For the token sequence $X^m \in \mathbb{R}^{N_m \times D}$ of modality $m$, the mask controls its visibility with the entire multimodal sequence $X \in \mathbb{R}^{(N_m+N_\bar{m}) \times D}$.

---

### Decision · Program_Chairs · 2025-05-01

**Decision:**

Accept (spotlight poster)

**Comment:**

The submission introduces MODA, a novel modular duplex attention mechanism designed to address the deficit disorder attention problem in multimodal large language models (MLLMs). Reviewers unanimously recognize the paper’s significant contributions, highlighting its clear novelty, comprehensive theoretical grounding, and extensive experimental validation.

While initial concerns regarding clarity and computational cost were raised, these were adequately addressed in the authors' rebuttal, resulting in reviewers increasing their ratings. Given the paper’s strong innovation, technical depth, and potential impact on multimodal AI research, the consensus among reviewers supports a clear acceptance. The AC agrees with the reviewers to accept the submission.